# SCALING UP AND STABILIZING DIFFERENTIABLE PLANNING WITH IMPLICIT DIFFERENTIATION

**Linfeng Zhao**[1*]**, Huazhe Xu**[23]**, Lawson L.S. Wong**[1]

[1]Khoury College of Computer Sciences, Northeastern University,
[2]Institute for Interdisciplinary Information Sciences, Tsinghua University,
[3]Shanghai Qi Zhi Institute

## ABSTRACT

Differentiable planning promises end-to-end differentiability and adaptivity. However, an issue prevents it from scaling up to larger-scale problems: they need to differentiate through forward iteration layers to compute gradients, which couples forward computation and backpropagation and needs to balance forward planner performance and computational cost of the backward pass. To alleviate this issue, we propose to differentiate through the Bellman fixed-point equation to decouple forward and backward passes for Value Iteration Network and its variants, which enables constant backward cost (in planning horizon) and flexible forward budget and helps scale up to large tasks. We study the convergence stability, scalability, and efficiency of the proposed implicit version of VIN and its variants and demonstrate their superiorities on a range of planning tasks: 2D navigation, visual navigation, and 2-DOF manipulation in configuration space and workspace.

## 1 INTRODUCTION

Planning is a crucial ability in artificial intelligence, robotics, and reinforcement learning (LaValle, 2006; Sutton & Barto, 2018). However, most planning algorithms require either a model that matches the true dynamics or a model learned from data. In contrast, differentiable planning (Tamar et al., 2016; Schrittwieser et al., 2019; Oh et al., 2017; Grimm et al., 2020; 2021) trains models and policies in an end-to-end manner. This approach allows learning a compact Markov Decision Process (MDP) and ensures that the learned value is equivalent to the original problem. For instance, differentiable planning can learn to play Atari games with minimal supervision (Oh et al., 2017).

However, differentiable planning faces scalability and convergence stability issues because it needs to differentiate through the planning computation. This process requires unrolling network layers iteratively to improve value estimates, especially for long-horizon planning problems. As a result, it leads to slower inference and inefficient and unstable gradient computation through multiple network layers. Therefore, this work addresses the question: *how can we scale up differentiatiable planning and keep the training efficient and stable?*

In this work, we focus on the bottleneck caused by *algorithmic differentiation*, which backpropagates gradients through layers and couples forward and backward passes and slows down inference and gradient computation. To address this issue, we propose implicit differentiable planning (IDP). IDP uses implicit differentiation to solve the fixed-point problem defined by the Bellman equations without unrolling network layers. Value Iteration Networks (VINs) (Tamar et al., 2016) use convolution networks to solve the fixed-point problem by embedding value iteration into its computation. We name it algorithmic differentiable planner, or *ADP* for short. We apply IDP to VIN-based planners such as GPPN (Lee et al., 2018) and SymVIN (Zhao et al., 2022). This implicit differentiation idea has also been recently studied in supervised learning (Bai et al., 2019; Winston & Kolter, 2021; Amos & Yarats, 2019; Amos & Kolter, 2019).

Using implicit differentiation in planning brings several benefits. It decouples forward and backward passes, so when the forward pass scales up to more iterations for long-horizon planning problems,

---

*Corresponding Author: Linfeng Zhao zhao.linf@northeastern.edu

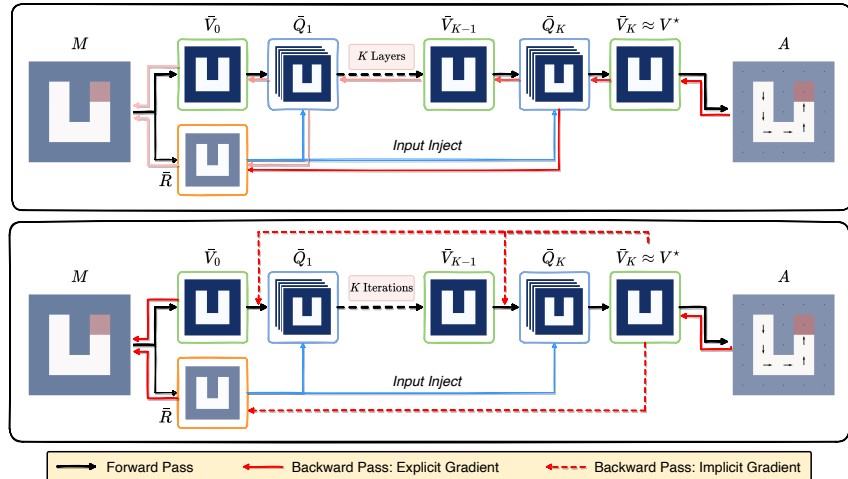

Figure 1: An overview of VIN, a planner with algorithmic differentiation, and ID-VIN, our proposed planner with implicit differentiation. Lighter colors for the backward pass with algorithmic differentiation (solid red arrows) indicate larger backpropagation depth. For backward passes with implicit differentiation, the dashed red arrows start from the solved equilibrium $V^\star$ and end at each forward layer (black arrows).

the backward pass can stay constant cost. It is also no longer constrained to differentiable forward solvers/planners, potentially allowing other non-differentiable operations in planning. It can potentially reuse intermediate computation from forward computation in the backward pass, which is infeasible for algorithmic differentiation. We focus on scaling up implicit differentiable planning to larger planning problems and stabilizing its convergence, and also experiment with different optimization techniques and setups. In our experiments on various tasks, the planners with implicit differentiation can train on larger tasks, plan with a longer horizon, use less (backward) time in training, converge more stably, and exhibit better performance compared to explicit counterparts. We summarize our contributions below:

- We apply implicit differentiation on VIN-based differentiable planning algorithms. This connects with deep equilibrium models (DEQ) (Bai et al., 2019) and prior work in both sides, including (Bai et al., 2021; Nikishin et al., 2021; Gehring et al., 2021).

- We propose a practical implicit differentiable planning pipeline and implement implicit differentiation version of VIN, as well as GPPN (Lee et al., 2018) and SymVIN (Zhao et al., 2022).

- We empirically study the convergence stability, scalability, and efficiency of the ADPs and proposed IDPs, on four planning tasks: 2D navigation, visual navigation, and 2 degrees of freedom (2-DOF) manipulation in configuration space and workspace.

## 2 RELATED WORK

**Differentiable Planning** In this paper, we use *differentiable planning* to refer to planning with neural networks, which can also be named *learning to plan* and may be viewed as a subclass of *integrating planning and learning* (Sutton & Barto, 2018). It is promising because it can be integrated into a larger differentiable system to form a closed loop. Grimm et al. (2020; 2021) propose to understand model-based planning algorithms from value equivalence perspective. Value iteration network (VIN) (Tamar et al., 2016) is a representative work that performs value iteration using convolution on lattice grids, and has been further extended (Niu et al., 2017; Lee et al., 2018; Chaplot et al., 2021; Deac et al., 2021) and Abstract VIN (Schleich et al., 2019). Other than using convolution network, the work on combining learning and planning includes (Oh et al., 2017; Karkus et al., 2017; Weber et al., 2018; Srinivas et al., 2018; Schrittwieser et al., 2019; Amos & Yarats, 2019; Wang & Ba, 2019; Guez et al., 2019; Hafner et al., 2020; Pong et al., 2018; Clavera et al., 2020).

**Implicit Differentiation** Beyond computing gradients by following the forward pass layer-by-layer, the gradients can also be computed with *implicit differentiation* to bypass differentiating through some advanced root-find solvers. This strategy has been used in a body of recent work

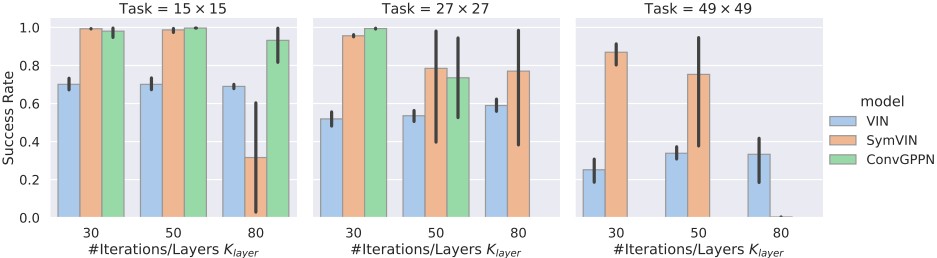

Figure 2: Demonstration of differentiable planners with algorithmic differentiation, which cannot scale up to large tasks or more iterations, due to coupled forward and backward pass.

(Chen et al., 2019; Bai et al., 2019; Amos & Kolter, 2019; Ghaoui et al., 2020). Particularly related, Bai et al. (2019) propose *Deep Equilibrium Models (DEQ)* that decouples the forward and backward pass and solve the backward pass iteratively also through a fixed-point system. Winston & Kolter (2021) study the convergence of fixed point iteration in a specific type of deep network. Amos & Kolter (2019) formalize optimization as a layer, and Amos & Yarats (2019) further apply the idea to iterative LQR. Gehring et al. (2021) theoretically study gradient dynamics of implicit parameterization of value function through the Bellman equation. Nikishin et al. (2021) similarly use implicit differentiation, while they explicitly solve the backward pass and only work on small-scale tasks because explicit solving is not scalable. Bacon et al. (2019) instead focus on a Lagrangian perspective. Our work is focused on scalability and convergence stability on differentiable planning, and experiments with challenging tasks in simulation to empirically justify the approach.

## 3 DIFFERENTIABLE PLANNING WITH ALGORITHMIC DIFFERENTIATION

**Background: Value Iteration Networks.** Value iteration is an instance of the dynamic programming (DP) method to solve Markov decision processes (MDPs). It iteratively applies the Bellman (optimality) operator until convergence, which is based on the following Bellman (optimality) equation: $Q(s, a) = R(s, a) + \gamma \sum_{s'} P(s'|s, a)V(s')$ and $V(s) = \max_a Q(s, a)$.

Tamar et al. (2016) used a convolution network to parameterize value iteration, named Value Iteration Network (VIN). VINs jointly learn and plan in a latent MDP on the 2D grid, which has the latent reward function $\bar{R} : \mathbb{Z}^2 \to \mathbb{R}^{|\mathcal{A}|}$ and has transition probability $\bar{P}$ represented as $W^V : \mathbb{Z}^2 \to \mathbb{R}^{|\mathcal{A}|}$, which only relies on differences between states. The value function is written as $\bar{V} : \mathbb{Z}^2 \to \mathbb{R}$ and $\bar{Q} : \mathbb{Z}^2 \to \mathbb{R}^{|\mathcal{A}|}$. Value iteration can be written as:

$$\bar{Q}_{\bar{a},i',j'}^{(k)} = \bar{R}_{\bar{a},i,j} + \sum_{i,j} W_{\bar{a},i,j}^V \bar{V}_{i'-i,j'-j}^{(k-1)}, \quad \bar{V}_{i,j}^{(k)} = \max_{\bar{a}} \bar{Q}_{\bar{a},i',j'}^{(k)}. \tag{1}$$

If we let $\boldsymbol{f}$ be a single application of the Bellman operator, Eq. 1 can be written as:

$$\bar{V}^{(k)} = \boldsymbol{f}(\bar{V}^{(k-1)}, \bar{R}, W^V) \equiv \max_a \bar{R}^a + W_{\bar{a}}^V \star \bar{V}^{(k-1)} \equiv \max_a \bar{R}^a + \texttt{Conv2D}(\bar{V}^{(k-1)}; W_{\bar{a}}^V) \tag{2}$$

where convolution $W_{\bar{a}}^V \star V$ is implemented as a 2D convolution layer $\texttt{Conv2D}$ with learnable weight $W^V$. For simplicity, we later use $\boldsymbol{\theta}$ to refer to network weights, and write each iteration as $\boldsymbol{v}_{k+1} = f(\boldsymbol{v}_k, \boldsymbol{r}, \boldsymbol{\theta})$, where $\boldsymbol{r}$ stands for reward map $\bar{R}$ and $\boldsymbol{v}_k$ for value map $\bar{V}^{(k)}$.

**Pitfall: Coupled forward and backward pass.** The forward computation of VIN iteratively applies the Bellman update $\boldsymbol{f}$. Thus, the optimization needs automatic differentiation: differentiating through multiple layers of forward iterations $\boldsymbol{f} \circ \boldsymbol{f} \circ \ldots \circ \boldsymbol{f}$. Using automatic differentiation for VIN has a major drawback: the forward computation of value iteration ("forward pass") and the computation of its gradients ("backward pass") are *coupled*. That is, if the planning horizon is enlarged, VINs would need larger number of iterations to propagate the value from goals to remaining positions. As used in VIN and GPPN (Tamar et al., 2016; Lee et al., 2018), it requires intensive memory usage to store every intermediate iterate $V^{(k)}$ to enable automatic differentiation.

To illustrate the effects, we show the performance of *ADP* in Figure 2, including three models with increasingly higher memory use and time cost: VIN, SymVIN (Zhao et al., 2022) and ConvGPPN (a modified version of GPPN (Lee et al., 2018)). They are trained on $15 \times 15$, $27 \times 27$, and $49 \times 49$ path

planning tasks. To solve larger mazes, each model consumes more memory while also needing more iterations (x-axes). However, since algorithmic differentiation requires backpropagating gradients layer by layer, with more iterations or larger tasks, some models either diverge or run out of memory (11GB limit). We present further experimental details and analyses in Section 5.2.

# 4 APPROACH: FROM *Explicit* TO *Implicit Differentiation* FOR PLANNING

This section introduces a strategy with *implicit differentiation* to resolve the issue by decoupling the forward and backward computation. We first derive implicit differentiation for VIN-based planners, then propose a pipeline for implementing those planners with implicit differentiation. We refer to them as *implicit differentiable planners* (IDP), in contrast to *algorithmic differentiable planners (ADP)* with algorithmic differentiation. We analyze the technical differences and similarities of IDPs vs. ADPs afterward.

## 4.1 IMPLICIT DIFFERENTIATION FOR VALUE ITERATION

We derive implicit differentiation for VIN and variants, where each layer $f$ is a step as in value iteration. Since the derivation does not rely on the concrete instantiation of $f$, we can freely replace $f$ from `Conv2D` with other types of layers. We will introduce these variants in the next subsection.

**Implicit differentiation.** A fixed-point problem can be solved iteratively by fixed-point iteration and other algorithms. However, as pointed out in (Bai et al., 2019), naively differentiating through the solver would require intensive memory usage, since it needs to store every intermediate iterate $V^{(k)}$ to enable automatic differentiation, as in (Tamar et al., 2016; Lee et al., 2018). As also used recently in (Bai et al., 2019; Nikishin et al., 2021; Gehring et al., 2021), another solution is to instead differentiate directly through the fixed point $z^*$ using the implicit function theorem and implicit differentiation. Then, we can skip all of this by decoupling forward (fixed-point iteration as the solver) and backward pass (differentiating through the solver).

We start with the fixed point equation $\boldsymbol{v}^\star = \boldsymbol{f}(\boldsymbol{v}^\star, \boldsymbol{r}, \boldsymbol{\theta})$ from the Bellman optimality equation. Below we use $\boldsymbol{x}$ to stand for either input $\boldsymbol{r}$ or $\boldsymbol{\theta}$. The implicit function theorem tells us that, under some mild conditions of the derivatives ($\boldsymbol{f}$ is continuously differentiable with non-singular Jacobian $\partial \boldsymbol{f}(\boldsymbol{v}, \boldsymbol{x})/\partial \boldsymbol{x}$), $\boldsymbol{v}^\star(\boldsymbol{x})$ is a differentiable function of $\boldsymbol{x}$ locally: $0 = f(\boldsymbol{v}^\star(\boldsymbol{x}), \boldsymbol{x})$.

For fixed point equation, we can assume the fixed point solution $\boldsymbol{v}^\star$ is obtained, thus this can be used to compute the partial derivative w.r.t. to any quantity (input, parameter, etc) $\partial \boldsymbol{v}^\star(\cdot)/\partial(\cdot)$. It avoids backpropagating gradients through the forward fixed-point iteration, which is computationally inefficient and requires considerable memory to store intermediate iteration variables. Additionally, it also allows to even use of non-differentiable operations in the forward pass.

Differentiating both sides of the equation $\boldsymbol{v}^\star = f(\boldsymbol{v}^\star, \boldsymbol{x})$ and applying the chain rule:

$$\frac{\partial \boldsymbol{v}^\star(\cdot)}{\partial(\cdot)} = \frac{\partial f(\boldsymbol{v}^\star(\cdot), \boldsymbol{x})}{\partial(\cdot)} = \frac{\partial f(\boldsymbol{v}^\star, \boldsymbol{x})}{\partial \boldsymbol{v}^\star} \frac{\boldsymbol{v}^\star(\cdot)}{\partial(\cdot)} + \frac{\partial f(\boldsymbol{v}^\star, \boldsymbol{x})}{\partial(\cdot)}, \tag{3}$$

where we use $(\cdot)$ to denote an arbitrary variable. Rearranging terms:

$$\frac{\partial \boldsymbol{v}^\star(\cdot)}{\partial(\cdot)} = \left( I - \frac{\partial f(\boldsymbol{v}^\star, \boldsymbol{x})}{\partial \boldsymbol{v}^\star} \right)^{-1} \frac{\partial f(\boldsymbol{v}^\star, \boldsymbol{x})}{\partial(\cdot)}. \tag{4}$$

**Solving backward pass.** To integrate into a deep learning framework for automatic differentiation, two quantities are needed: VJP (vector-Jacobian product) and JVP (Jacobian-vector product) (Gilbert, 1992). Nevertheless, the computation of the inverse term $(I - \partial f(\boldsymbol{v}^\star, \boldsymbol{x})/\partial \boldsymbol{v}^\star)^{-1}$ can be a major bottleneck due to its dimension ($O(d^2)$ or $O(m^4)$, where $d = m^2$ is the matrix width and $m$ is the map size) (Bai et al., 2019; 2021). Additionally, when applied to VINs, we concatenate a policy layer that maps the final equilibrium $\boldsymbol{v}^\star \in \mathbb{R}^{m \times m}$ to action logits $\mathbb{R}^{m \times m}$ and compute the cross-entropy loss $\ell(\cdot)$ (Tamar et al., 2016). Thus, the derivative of loss is:

$$\frac{\partial \ell}{\partial(\cdot)} = \frac{\partial \ell}{\partial \boldsymbol{v}^\star} \frac{\partial \boldsymbol{v}^\star(\cdot)}{\partial(\cdot)} = \frac{\partial \ell}{\partial \boldsymbol{v}^\star} \left( I - \frac{\partial f(\boldsymbol{v}^\star, \boldsymbol{x})}{\partial \boldsymbol{v}^\star} \right)^{-1} \frac{\partial f(\boldsymbol{v}^\star, \boldsymbol{x})}{\partial(\cdot)}, \tag{5}$$

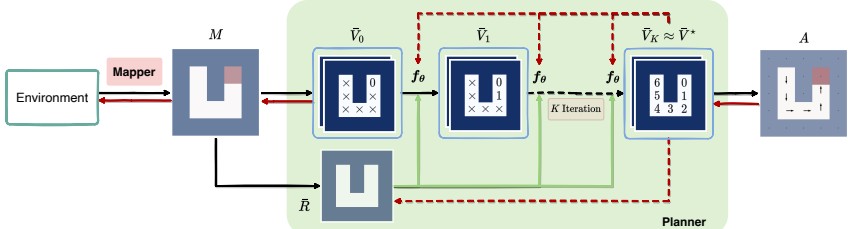

Figure 3: The proposed pipeline of implicit differentiable planning by using implicit differentiation.

Defining $\boldsymbol{w}$ as follows forms a linear fixed-point system (Bai et al., 2019):

$$\boldsymbol{w}^\top \triangleq \frac{\partial \ell}{\partial \boldsymbol{v}^\star} \left( I - \frac{\partial f(\boldsymbol{v}^\star, \boldsymbol{x})}{\partial \boldsymbol{v}^\star} \right)^{-1} ; \quad \boldsymbol{w}^\top = \boldsymbol{w}^\top \frac{\partial f(\boldsymbol{v}^\star, \boldsymbol{x})}{\partial \boldsymbol{v}^\star} + \frac{\partial \ell}{\partial \boldsymbol{v}^\star} . \quad (6)$$

This backward pass fixed-point equation can also be solved by a generic fixed-point solver or root finder. Then, we can substitute the solution $\boldsymbol{w}$ back: $\partial \ell / \partial (\cdot) = \boldsymbol{w}^\top \partial f(\boldsymbol{v}^\star, \boldsymbol{x}) / \partial (\cdot)$. The computation is then purely based on VJP and JVP. In summary, an IDP needs to solve both the (nonlinear) *forward* fixed-point system and the (linear) *backward* fixed-point system, as in DEQ.

## 4.2 A Pipeline of Implicit Differentiable Planning

We can derive variants of VIN using implicit differentiation by abstracting out the implementation of value iteration layer. In this section, we propose a generic implicit planning pipeline to extend our approach to Gated Path Planning Networks (GPPN) (Lee et al., 2018) and Symmetric VIN (SymVIN) (Zhao et al., 2022). Spatial Planning Transformers (SPT) (Chaplot et al., 2021) also fits into the pipeline, but it performs less well, as discussed in Section C.

Figure 3 shows the general pipeline, where the network layer $\boldsymbol{f}_\theta$ can be replaced by any single layer that is capable of iterating values. The pipeline follows VIN and GPPN, where for 2D path planning a map $\mathbb{Z}^2 \to \{0, 1\}$ is provided, and the planners' output actions (their logits) for each position $\mathbb{Z}^2 \to \mathbb{R}^{|\mathcal{A}|}$. All these tasks can be represented as taking a form of map "signal" over grid $\mathbb{Z}^2$, as previously been done (Zhao et al., 2022; Chaplot et al., 2021).

**Planner instantiations.** We now introduce the instantiations of implicit planners one by one. We focus on the value iteration part (omit map input and action output), and all planners follow the form $\bar{V}^{(k+1)} = f_{\boldsymbol{\theta}}(\bar{V}^{(k)}, \bar{R})$ (bars omitted later). There are two design principles: (1) input inject ($\bar{R}$ must be input, as input $\boldsymbol{x}$) and (2) weight-tied ($\boldsymbol{\theta}$ is shared across layers $f$), as also used in DEQ (Bai et al., 2019). Specifically, the purpose of input inject is that fixed-point solution $\bar{V}^\star$ does not depend on initialization $\bar{V}^{(0)}$, so we must pass information of the map through input inject (by $\bar{R}$).

*(i)* **ID-VIN** uses regular 2D translation-equivariant convolution layer, where $f(V, R) = \max_a R^a +$ `Conv2D`$(V; \theta)$. *(ii)* **ID-SymVIN** aims to integrate symmetry into planning and uses equivariant $E(2)$-steerable CNN (Weiler & Cesa, 2021). It has similar form to VIN and just replaces `Conv2D` with `SteerableConv`, thus the form is $f(V, R) = \max_a R^a +$ `SteerableConv`$(V; \theta)$.

*(iii)* **ID-ConvGPPN** is based on our modified version of GPPN (Kong, 2022; Zhao et al., 2022), where we (1) use GRU since it has a single input with form $z' = $ `GRU`$(z, x)$ and is easier to integrate into our current form, (2) replace all fully connected layers with convolution layers, and (3) inject $R$ to every step. The result is that every layer is a ConvGRU, instead of LSTM in GPPN: $f(V, R) = $ `ConvGRU`$(V, R; \theta)$. Note that the GPPN variants do not have `max` in each iteration anymore and directly take reward $R$ to the recurrent cell (Lee et al., 2018).

**Mapper Layer.** We can handle tasks with more challenging input, such as *visual navigation* and *workspace manipulation* (Lee et al., 2018; Chaplot et al., 2021; Zhao et al., 2022), by learning an additional mapping network (*mapper*) to first map the input to a 2D map. Further details about environments and mapper implementation are deferred to Section 5.1 and Section C.

**Optimization.** We build upon the deep equilibrium model (DEQ) and relevant work (Bai et al., 2019; 2020; 2021), which includes several effective techniques. By representing the implementation of value iteration as fixed-point solving, we have the flexibility to use any fixed-point or root solver

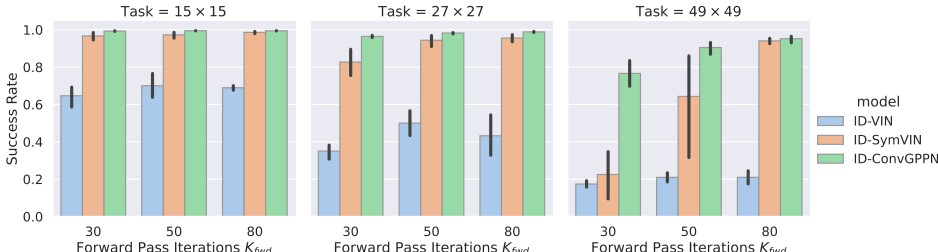

Figure 4: Performance of implicit differentiable planners on 2D navigation tasks. Since the forward and backward pass are decoupled, the IDPs can keep consistent runtime and memory cost for backward pass while have forward pass of value iteration converged for large tasks and iterations.

for $f(\boldsymbol{v}, \boldsymbol{\theta}) = \boldsymbol{v}$ or $\boldsymbol{v} - f(\boldsymbol{v}, \boldsymbol{\theta}) = 0$. A straightforward solver is forward iteration, as used in VIN's feedforward network (Tamar et al., 2016). However, recent work has employed Anderson's or Broyden's method (Bai et al., 2019; 2020). In our experiments, we compare the use of forward iteration and the Anderson solver in both forward and backward passes. Notably, the SymVIN architecture requires the entire forward pass to be equivariant, so extra attention is necessary when designing the forward solver. Further details and results can be found in Sections C and E.

### 4.3 Implicit vs. Explicit Differentiable Planners

**Underlying computational similarity.** The gradient computation is done by automatic differentiation (Gilbert, 1992). For *algorithmic differentiation*, the gradients are computed through direct backpropagation and the implementation is also based on efficiently computing vector-Jacobian product (VJP) and Jacobian-vector product (JVP) (Bai et al., 2019). Christianson (1994) studied automatic differentiation for implicit differentiation of fixed-point system. The only difference is the number of operations required and that implicit differentiation is based on the Jacobian at the equilibrium. We derive the connection in Section B.1.

**Comparison: Implicit vs. algorithmic differentiation.** In Figure 4, we compare the performance of three IEPs with different numbers of iterations in the forward pass. Unlike the ADPs shown in Figure 2, our IDPs converge stably with larger forward-pass iterations (horizontal axis), while ADPs sometimes diverge on large iterations. Note that we should not directly compare IDPs and ADPs with the same number of forward iterations since they refer to different things: IDPs compute an equilibrium and use it for backward pass (more forward iterations would be better), while for ADPs # iterations = # layers (more iterations/layers leads to instability). Further analyses in Section 5.2.

**Tradeoff: The quality of equilibrium.** Theoretically, IDPs have a constant cost of the backward pass with respect to the forward planning horizon. However, the *backward pass* requires the Jacobian of the final equilibrium $\boldsymbol{v}^\star \approx \boldsymbol{v}_K$ from the forward pass. If the equilibrium $\boldsymbol{v}^\star$ is not solved reasonably well, the backward pass in Eq. 6 would iterate based on an inaccurate Jacobian $\partial f(\boldsymbol{v}^\star, x)/\partial \boldsymbol{v}^\star$, which would cause poor performance. In contrast, because ADPs compute exact gradients by backpropagation through layers, they do not suffer from this issue. Additionally, fewer iterations (layers) would also alleviate their convergence issues.

Empirically, with fewer *forward iterations* (e.g., 10), we observe a performance drop from IDPs. Although ADPs also perform worse with fewer layers, they have a smaller drop compared to IDPs. However, when scaling up to more forward iterations $K_{\mathrm{fwd}}$ (e.g. $\geq 30$ iterations, which are necessary for maps $\geq 27 \times 27$) IDPs may solve the equilibrium well enough and are more favorable because of their efficient backward pass. Moreover, we find that using around $K_{\mathrm{bwd}} \approx 15$ iterations for backward pass works well enough consistently across different map sizes ($15 \times 15$ through $49 \times 49$). Since both algorithmic and implicit differentiation use a similar amount of vector-matrix products, using more than $K_{\mathrm{layer}} \geq 15 \approx K_{\mathrm{bwd}}$ would consume more resources and favor IDPs.

## 5 Empirical Analysis

We present more results on convergence, scalability, generalization, and efficiency, which extends the study in previous sections on comparing implicit vs. algorithmic differentiable planners.

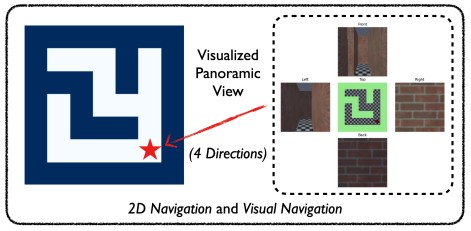 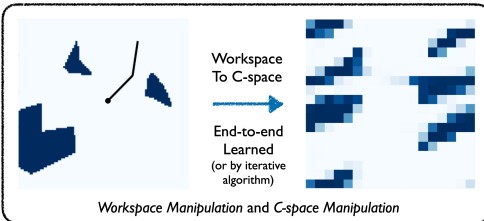

Figure 5: **(Left)** We randomly generate occupancy grid $\mathbb{Z}^2 \to \{0, 1\}$ for **2D navigation**. For **visual navigation**, each position provides $32 \times 32 \times 3$ egocentric panoramic RGB images in 4 directions for each location $\mathbb{Z}^2 \to \mathbb{R}^{4 \times 32 \times 32 \times 3}$. One location is visualized. **(Right)** The top-down view (left) is the *workspace* of a 2-DOF manipulation task. For **workspace manipulation**, it is converted by a mapper layer to configuration space, shown in the right subfigure. For **C-space manipulation**, a ground-truth C-space is provided to planners.

## 5.1 ENVIRONMENTS AND SETUP

**Environments and datasets.** We run implicit and algorithmic differentiable planners on four types of tasks: (1) **2D navigation**, (2) **visual navigation**, (3) 2 degrees of freedom (2-DOF) **configuration space manipulation**, and (4) **2-DOF workspace manipulation**. These tasks require planning on either *given* (2D navigation and 2-DOF configuration-space manipulation) or *learned* maps (visual navigation and 2-DOF workspace manipulation), where the maps are 2D regular grid as in prior work (Tamar et al., 2016; Lee et al., 2018; Chaplot et al., 2021). To learn maps, a planner needs to jointly learn a mapper that converts egocentric panoramic images (visual navigation) or workspace states (workspace manipulation) into a 2D grid. We follow the setup in (Lee et al., 2018; Chaplot et al., 2021) and further discuss in Section D. In both cases, we randomly generate training, validation and test data of $10K/2K/2K$ maps for all map sizes. For all maps, the action space is to move in 4 $\circlearrowleft$ directions: $\mathcal{A} = (\texttt{north}, \texttt{west}, \texttt{south}, \texttt{east})$.

**Training and evaluation.** We report success rate and training curves over 5 seeds. The training process (on given maps) follows (Tamar et al., 2016; Lee et al., 2018; Zhao et al., 2022), where we train 60 epochs with batch size 32, and use kernel size $F = 3$ by default. We use RTX 2080 Ti cards with 11GB memory for training, thus we use 11GB as the memory limit for all models.

## 5.2 CONVERGENCE AND SCALABILITY

In the previous sections with Figure 2 and 4, we have presented quantitive analysis of IDPs and ADPs in terms of convergence with more iterations and scalability on larger tasks. Here, we provide the detailed setup of the experiment and put the attention more on the qualitative side.

**Setup.** We train all models on 2D maze navigation tasks with map sizes $15 \times 15$, $27 \times 27$, and $49 \times 49$. We use $K_{\text{layer}} = 30, 50, 80$ iterations for *ADPs*, which is effectively the number of layers in their networks. Correspondingly, for *IDPs*, we choose to use *forward iteration* solver for forward pass and Anderson solver for backward pass. We fix the number of iterations of backward solver as $K_{\text{bwd}} = 15$ and of forward solver as $K_{\text{fwd}} = 30, 50, 80$.

**Results.** We examine results by algorithm and focus on their trend with iteration number (x-axis) and map size (column), not just the absolute numbers. The conclusion already mentioned in the above section: Beyond an intermediate iteration number (around 30-50), IDPs are more favorable because of scalability and computational cost. We present other analyses here.

We start from ConvGPPN and ID-ConvGPPN, which perform the best in ADPs and IDPs class, respectively. They also have the most number of parameters and use greatest time because of the gates in ConvGRU units. As shown in Figure 2, this also caused two issues of ConvGPPN: scalability to larger maps/iterations (out of memory for $27 \times 27$ 80 iterations and $49 \times 49$ 50 and 80 iterations), and also convergence stability (e.g. $27 \times 27$ 50 iterations).

For SymVIN and ID-SymVIN, they replace `Conv2D` with `SteerableConv`, with computational cost slightly higher than VIN and much lower than ConvGPPN. Thus, they can successfully run on all tasks and iteration numbers. However, we find that explicit SymVIN may diverge due to bad initialization, and this is more severe if the network is deeper (more iterations), as in Figure 2's 50 and

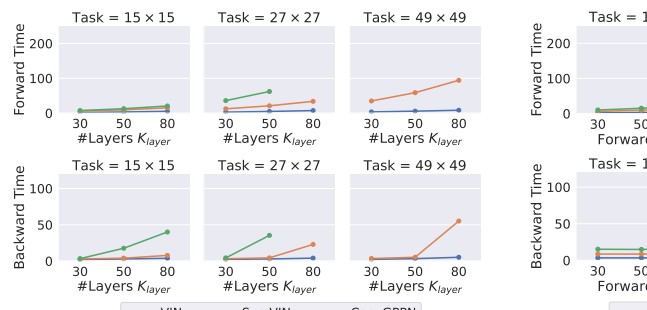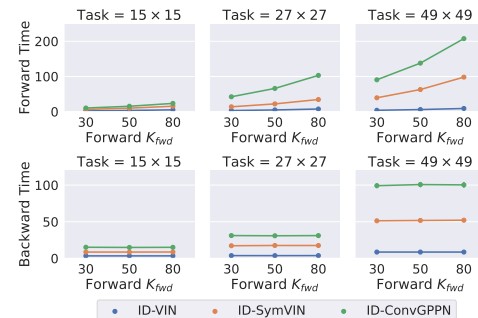

Figure 6: The runtime (in seconds) on 2D navigation tasks with size $15 \times 15$, $27 \times 27$, and $49 \times 49$, averaged over 5 seeds. The ADPs are on the *left* six figures and the IDPs on the *right*. *Missing dots* are due to out of memory caused by algorithmic differentiation. The *upper* row is for forward pass runtime, and the *lower* row is for backward runtime. The horizontal axes mean differently: (1) ADPs: the number of layers $K_{\text{layer}}$, also number of iterations, and (2) IDPs: the forward pass iterations $K_{\text{fwd}}$.

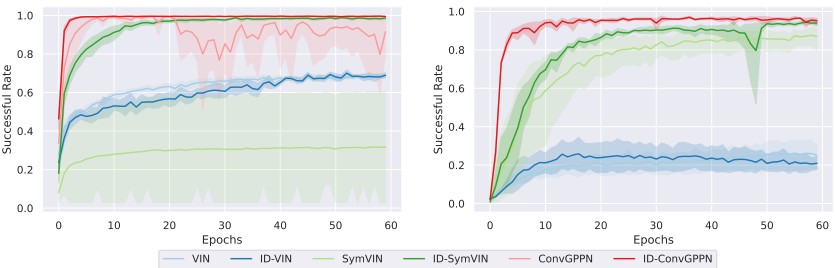

Figure 7: **(Left)** Training curves on 2D maze navigation $15 \times 15$ maps, with 80 layers for ADPs and 80 iterations for IDPs. **(Right)** Training curves on $49 \times 49$ maps, with 30 layers for ADPs (due to scalability issue) and 80 iterations for IDPs.

80 iterations. Nevertheless, ID-SymVIN alleviates this issue, since implicit differentiable planning decouples forward and backward pass, so the gradient computation is not affected by forward pass.

Furthermore, VIN and ID-VIN are surprisingly less affected by the number of iterations and problem scale. We find their forward passes tends to converge faster to reasonable equilibria, thus further increasing iteration does not help, nor break convergence as long as memory is sufficient.

**Forward and backward runtime.** We visualize the runtime of IDPs and ADPs in Figure 6. For IDPs, we use the forward-iteration solver for the forward pass and Anderson solver for the backward pass. Note that in the bottom left, we intentionally plot backward runtime vs. forward pass iterations. This emphasizes that IDPs decouple forward and backward passes because the backward runtime does not rely on forward pass iterations. Instead, for ADPs, value iteration is done by network layers, thus the backward pass is coupled: the runtime increases with depth and some runs failed due to limited memory (11GB, see missing dots).

Therefore, this set of figures shows better scalability of IDPs (no missing dots – out of memory – and constant backward time). In terms of absolute time, the forward runtime of IDPs when using the *forward solver* is comparable with successful ADPs.

## 5.3 TRAINING PERFORMANCE

**Setup.** Beyond evaluating generalization to novel maps, we compare their training efficiency with learning curves. Each learning curve is aggregated over 5 seeds, which are from the models in the above section. The learning curves are for all planners on $15 \times 15$ maps (Figure 7 left) and $49 \times 49$ maps (Figure 7 right, 30 layers for ADPs – due to scalability issue – and 80 iterations for IDPs).

**Results.** On $15 \times 15$ maps, we show $K_{\text{layer}} = 80$ layers for ADPs and $K_{\text{fwd}} = 80$ iterations for IDPs. ID-ConvGPPN performs the best and is much more stable than its ADP counterpart ConvGPPN. ID-SymVIN learns reliably, while SymVIN fails to converge due to instability from 80 layers. ID-VIN and VIN are comparable throughout training. On $49 \times 49$ maps, we visualize $K_{\text{layer}} = 30$ layers

for ADPs (due to their limited scalability) and $K_{\text{fwd}} = 80$ iterations for IDPs. ConvGPPN cannot run at all even for only 30 layers, while ID-ConvGPPN still reaches a near-perfect success rate. ID-SymVIN learns slightly better than SymVIN and reaches higher asymptotic performance. ID-VIN has a similar trend to VIN, but performs worse overall due to the complexity of the task.

## 5.4 PERFORMANCE ON MORE TASKS

Table 1: Averaged test success rate (%) over 5 seeds for using 10K/2K/2K dataset on the rest of 3 types of tasks. We highlight entry with *italic* for runs with at least one diverged trial (any success rate $< 20\%$).

| Type | Methods | $18 \times 18$ Mani. | $36 \times 36$ Mani. | Workspace Mani. | Visual Nav. |
|---|---|---|---|---|---|
| Explicit | VIN | $89.65_{\pm 7.97}$ | $74.75_{\pm 8.18}$ | $80.98_{\pm 3.84}$ | $66.11_{\pm 8.91}$ |
| | SymVIN | $55.15_{\pm 49.54}$ | $65.72_{\pm 47.11}$ | $82.17_{\pm 24.72}$ | $96.04_{\pm 4.24}$ |
| | ConvGPPN | $79.71_{\pm 20.71}$ | $70.55_{\pm 36.13}$ | $70.23_{\pm 19.44}$ | $81.76_{\pm 31.04}$ |
| **Implicit** (ours) | ID-VIN | $80.53_{\pm 6.98}$ | $56.27_{\pm 20.92}$ | $77.17_{\pm 7.24}$ | $62.53_{\pm 15.93}$ |
| | ID-SymVIN | $99.63_{\pm 0.08}$ | $98.53_{\pm 1.42}$ | $87.60_{\pm 24.11}$ | $86.41_{\pm 30.34}$ |
| | ID-ConvGPPN | $97.28_{\pm 0.74}$ | $93.60_{\pm 1.68}$ | $92.60_{\pm 1.83}$ | $98.91_{\pm 0.34}$ |

**Setup.** We run all planners on the other three challenging tasks. For **visual navigation**, we randomly generate 10K/2K/2K maps using the same strategy as 2D navigation and then render four egocentric panoramic views for each location from produced 3D environments with *Gym-MiniWorld* (Chevalier-Boisvert, 2018). For **configuration-space manipulation** and **workspace manipulation**, we randomly generate 10K/2K/2K tasks with 0 to 5 obstacles in workspace. In configuration-space manipulation, we manually convert each task into a $18 \times 18$ or $36 \times 36$ map ($20°$ or $10°$ per bin). The workspace task additionally needs a mapper network to convert the $96 \times 96$ workspace (image of obstacles) to an $18 \times 18$ 2-DOF configuration space (2D occupancy grid). We provide additional details in the Section D.

**Results.** In Table 1, due to space limitations, we average over $K_{\text{layer}} = 30, 50, 80$ for ADPs and $K_{\text{fwd}} = 30, 50, 80$ for IDPs. For each task, we present the mean and standard deviation over 5 seeds times three hyperparameters and provide the separated results to Section E. We italicize entries for runs with at least one diverged trial (any success rate $< 20\%$).

Generally, IDPs perform much more stably. On $18 \times 18$ or $36 \times 36$ configuration-space manipulation, ID-SymVIN and ID-ConvGPPN reach almost perfect results, while ID-VIN has diverged runs on $36 \times 36$ (marked in *italic*). SymVIN and ConvGPPN are more unstable, while VIN even outperforms them and is also better than ID-VIN. On $18 \times 18$ workspace manipulation, because of the difficulty of jointly learning maps and potentially planning on inaccurate maps, most numbers are worse than in configuration-space. ID-ConvGPPN still performs the best, and other methods are comparable. For $15 \times 15$ visual navigation, it needs to learn a mapper from panoramic images and is more challenging. ID-ConvGPPN is still the best. ID-SymVIN exhibits some failed runs and gets underperformed by SymVIN in these seeds, and ID-VIN is comparable with VIN.

Across all tasks, the results confirm the superiority of scalability and convergence stability of IDPs and demonstrate the ability of jointly training mappers (with algorithmic differentiation for this layer) even when using implicit differentiation for planners.

## 6 CONCLUSION

This work studies how VIN-based differentiable planners can be improved from an implicit-function perspective: using implicit differentiation to solve the equilibrium imposed by the Bellman equation. We develop a practical pipeline for implicit differentiable planning and propose implicit versions of VIN, SymVIN, and ConvGPPN, which is comparable to or outperforms their explicit counterparts. We find that implicit differentiable planners (IDPs) can scale up to longer planning-horizon tasks and larger iterations. In summary, IDPs are favorable for these cases to ADPs for several reasons: (1) better performance mainly due to stability, (2) can scale up while some ADPs fail due to memory limit, (3) less computation time. On the contrary, if using too few iterations, the equilibrium may be poorly solved, and ADPs should be used instead. While we focus on value iteration, the idea of implicit differentiation is general and applicable beyond path planning, such as in continuous control where Neural ODEs can be deployed to solving ODEs or PDEs.

## 7 ACKNOWLEDGEMENT

This work was supported by NSF Grant #2107256. We also thank Clement Gehring and Lingzhi Kong for helpful discussions and anonymous reviewers for useful feedback.

## 8 REPRODUCIBILITY STATEMENT

We provide an appendix with extended details of implementation in Section D, experiment details in Section D, and additional results in Section E. We plan to open-source our code next.

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

## CONTENTS

## A  OUTLINE

We provide additional discussion, extended details of implementation and experiments, and additional results in the appendix. The table of content is available above.

## B  EXTENDED DISCUSSION

### B.1  COMPUTATIONAL SIMILARITY

In value iteration, we iteratively apply $\boldsymbol{f}$ until convergence: $\boldsymbol{f}(\boldsymbol{v}_k, \boldsymbol{r}, \boldsymbol{\theta}) = \boldsymbol{v}_{k+1}$, then the outer loop optimizes one step on updating the model $\boldsymbol{\theta}$. We can generalize to time-varying optimization problem with equality constraint (Bacon et al., 2019): $\boldsymbol{f}_t(\boldsymbol{v}_t, \boldsymbol{r}, \boldsymbol{\theta}_t) = \boldsymbol{v}_{t+1}$, and we assume a mapping $\phi_t$ gives the inner optimization of VI: $\phi_t = \boldsymbol{f}_{t-1} \circ \ldots \circ \boldsymbol{f}_0, \phi_t : \boldsymbol{\theta}_{0:t} \mapsto \boldsymbol{v}_t$. We

$$\frac{\partial \ell}{\partial \boldsymbol{\theta}_t} = \frac{\partial \ell}{\partial \boldsymbol{v}_T} \frac{\partial \boldsymbol{\phi}_T}{\partial \boldsymbol{\theta}_t} = \frac{\partial \ell}{\partial \boldsymbol{v}_T} \lambda_T^\top \tag{7}$$

$$= \frac{\partial \ell}{\partial \boldsymbol{v}_T} \frac{\partial \boldsymbol{f}_{T-1}(\boldsymbol{v}_{T-1}, \boldsymbol{\theta}_{T-1})}{\partial \boldsymbol{v}_{T-1}} \frac{\partial \boldsymbol{\phi}_{T-1}(\boldsymbol{\theta}_{T-1})}{\partial \boldsymbol{\theta}_t} \tag{8}$$

$$= \frac{\partial \ell}{\partial \boldsymbol{v}_T} \frac{\partial \boldsymbol{f}_{T-1}}{\partial \boldsymbol{v}_{T-1}} \frac{\partial \boldsymbol{f}_{T-2}}{\partial \boldsymbol{v}_{T-2}} \ldots \frac{\partial \boldsymbol{\phi}_t(\boldsymbol{\theta}_t)}{\partial \boldsymbol{\theta}_t} \tag{9}$$

The recursion is then given by

$$\lambda_T^\top = \frac{\partial \boldsymbol{\phi}_T(\theta_T)}{\partial \theta_T}, \quad \lambda_{k+1}^\top = \frac{\partial \boldsymbol{f}_k}{\partial \boldsymbol{v}_k} \cdot \lambda_k^\top. \tag{10}$$

The vector $\lambda_t^\top$ is the adjoint vector, or costate, in control theory. The recursive update is the adjoint equation. The computation is exactly the vector-Jacobian product used in automatic differentiation (Griewank & Walther, 2008).

This shows close connection between Equation 6 on the fixed-point solving for backward pass of implicit differentiation and 10 on the recursive computation for algorithmic differentiation.

### B.2 MORE DISCUSSIONS

**Divergence of fixed-point iteration.** The term "divergence" can be confusing, so we'll clarify its meaning. In value iteration, the goal of the forward pass is to find a fixed point by iteratively applying the Bellman optimality operator. For an MDP with a given dynamics model, Bellman operators induced from the model and arbitrary policy are guaranteed to be a contraction mapping, so iterative methods will converge to a unique fixed point.

However, when the entire network is learned end-to-end, and the transition weights are also learned, the property of Bellman operators is not guaranteed. When the planning horizon increases, this issue becomes more severe since it can cause the iteration to move farther from the fixed point.

When we talk about divergence, we mainly refer to the fixed-point iteration in the forward pass. This iterative process does not involve gradient computation, but is purely a property of VIN and its variants.

It's worth noting that implicit differentiable planners introduce a fixed-point iteration for the backward pass, but empirically, they do not have a divergence issue.

**Performance of implicit vs. algorithmic differentiation.** For implicit differentiable planners, implicit differentiation itself does not guarantee to result in better performance. Implicit differentiation and algorithmic differentiation have equivalent asymptotic performance and are not expected to perform differently if run for a long enough time. However, the advantage of implicit differentiable planners is that they scale up better and can run with fewer resources, while the algorithmic differentiation ones cannot keep up with the scale.

**Relationship with DEQ (Bai et al., 2019).** The focus of our paper is to address the specific problem of scaling up and stabilizing differentiable planning algorithms, particularly for value iteration-based fixed-point solving methods, such as VIN and its variants. Our approach is problem-driven, rather than method-driven like DEQ. While our work does draw on some prior techniques and models, we introduce new insights and methods that are specific to our problem domain.

Technical contributions: Although our paper does build on some prior work, we make several significant technical contributions that are specific to VIN and variants. For example, we introduce the use of implicit differentiation, which has a direct correspondence to the differentiable planning algorithm and allows for more scalable and interpretable solutions. We also explore specific tuning and techniques that are needed to address the unique challenges of value iteration-based methods, such as the need for precise value function calculations and the limitations of using fixed-point iteration in the planning process.

While DEQ has inspired some of our ideas and approaches, we also highlight some key differences between our work and DEQ. For instance, DEQ is designed for supervised learning and does not require the same level of precision and stability as value iteration-based methods. Additionally, DEQ requires weight-tying and input injection, which are already present in vanilla VIN.

In conclusion, while our paper draws on some prior work and techniques, we introduce new insights and methods that are specific to VIN and variants. We address the challenges of scaling up and stabilizing value iteration-based methods, and introduce the use of implicit differentiation, which offers more scalable and interpretable solutions for these types of algorithms.

## C  Implementation Details

### C.1  Implementation of ID-SPT

Beyond VIN, SymVIN and ConvGPPN, we also tried implementing an implicit differentiation version of Spatial Planning Transformers (SPT) (Chaplot et al., 2021). However, we find the reimplementation of SPT and also the implicit version both do not work well enough. We first introduce how we implemented it and discuss our hypotheses on its failure.

**Implementation. ID-SPT** is based on a Transformer architecture, SPT. SPT is proposed to facilitate global value propagation using the global receptive field of self-attention layers in Transformers. Different from other variants, SPT uses heavier global self-attention layer and does not scale up the number of layers with map size, although the number of weights increases quadratically with size. We also implement an implicit version by using individual self-attention layer, where $f(V) = \texttt{SelfAtt}(V; \theta)$. Even SPT fits into our pipeline, we empirically find SPT behaves unlike other planners since it does not inject reward $R$ as input.

**Discussion of performance.** In our experiments, we find the modified ID-SPT cannot outperform SPT.

Since SPT uses multiple Transformer (self-attention) layers, it is computationally expensive. Thus, we use much smaller number of iterations for ID-SPT: $K = 3, 5, 10, 15$, because the original paper uses $K = 5$ layers across all map sizes.

However, we plot the convergence curve for its forward and backward pass. We find that the forward pass can only convert to around $10^0 = 1$ to $10^{-1}$ level (relative residual) and cannot further decrease, while other planners have their forward pass converged around at least $10^{-2}$. We find this may affect the backward pass, as the Jacobian at the solved equilibrium is used in solving the backward fixed-point iteration.

Considering these, we think the reason might come from the fact that SPT uses Transformer layers, which is too expressive and tends to learn an arbitrary output as it needs.

### C.2  Optimization of Implicit Differentiable Planners

To optimize the performance of IDPs, we also implemented other techniques. We tried *Jacobian regularization* from (Bai et al., 2021), which estimates the Jacobian $\left\| \frac{\partial f_\theta(\boldsymbol{v}^\star; \mathbf{x})}{\partial \boldsymbol{v}^\star} \right\|_F$ at the equilibrium.

We experiment it on ID-VIN, since other methods have more memory use and Jacobian loss would cause out of memory. However, it does not perform as we expected and rather decreases the success rate. We provide additional results on this in the later result section.

## D  Experiment Details

### D.1  Building Mapper Networks

**For visual navigation.**  For navigation, we follow the setting in GPPN (Lee et al., 2018). The input is $m \times m$ panoramic egocentric RGB images in 4 directions of resolution $32 \times 32 \times 3$, which forms a tensor of $m \times m \times 4 \times 32 \times 32 \times 3$. A mapper network converts every image into a 256-dimensional embedding and results in a tensor in shape $m \times m \times 4 \times 256$ and then predicts map layout $m \times m \times 1$.

For the first image encoding part, we use a CNN with the first layer of 32 filters of size $8 \times 8$ and stride of $4 \times 4$, and the second layer with 64 filters of size $4 \times 4$ and stride of $2 \times 2$, with a final linear layer of size 256.

In the second obstacle prediction part, the first layer has 64 filters and the second layer has 1 filter, all with filter size $3 \times 3$ and stride $1 \times 1$.

**For workspace manipulation.**  For **workspace manipulation**, we use U-net (Ronneberger et al., 2015) with residual-connection (He et al., 2015) as a mapper, see Figure 8. The input is $96 \times$

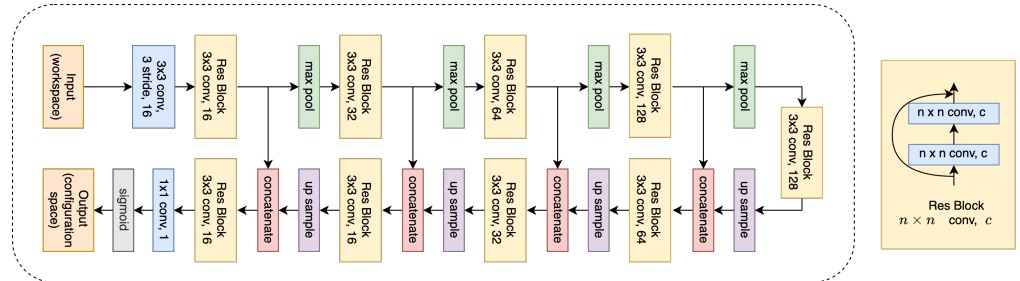

Figure 8: The U-net architecture we used as manipulation mapper.

96 top-down occupancy grid of the workspace with obstacles, and the target is to output $18 \times 18$ configuration space as the maps for planning.

During training, we pre-train the mapper and the planner separately for 15 epochs. Where the mapper takes manipulator workspace and outputs configuration space. The mapper is trained to minimize the binary cross entropy between output and ground truth configurations space. The planner is trained in the same way as using given maps. After pre-training, we switch the input to the planner from ground truth configuration space to the one from the mapper. During testing, we follow the pipeline in Chaplot et al. (2021) that the mapper-planner only have access to the manipulator workspace.

## D.2 Training Setup

We try to mimic the setup in VIN and GPPN (Lee et al., 2018).

For non-SymPlan related parameters, we use learning rate of $10^{-3}$, batch size of 32 if possible (GPPN variants need smaller), RMSprop optimizer.

For SymPlan parameters, we use 150 hidden channels (or 150 *trivial* representations for SymPlan methods) to process the input map. We use 100 hidden channels for Q-value for VIN (or 100 *regular* representations for SymVIN), and use 40 hidden channels for Q-value for GPPN and ConvGPPN (or 40 *regular* representations for SymGPPN on $15 \times 15$, and 20 for larger maps because of memory constraint).

## E    Additional Results

### E.1    2D Navigation Training on $75 \times 75$ Maps

We include comparison of differentiable planners with implicit differentiation or algorithmic differentiation on even larger maps: $75 \times 75$. Every error bar contains 3 to 5 seeds.

Note that this has not been done in any of prior work along this line, including VIN, SPT, SymVIN and more. This shows better scalability of the implicit differentiable planners. Prior work GPPN uses only $28 \times 28$ and SPT uses $50 \times 50$ (which mainly emphasizes long-term planning) for both training and evaluation. Note that SymVIN also only uses $50 \times 50$ for training, while use up to $100 \times 100$ for generalization. We also did the same experiment, available in Section E.2.

Figure 9 shows the performance of all implicit and explicit differentiable planners on the $75 \times 75$ tasks. We train explicit differentiable planners with $K_{\text{layer}} = 30, 50, 80, 120$ layers, and our implicit differentiable planners with max $K_{\text{fwd}} = 30, 50, 80, 120$ forward pass iterations and $K_{\text{bwd}} = 15$ backward pass iterations

For the explicit side, ConvGPPN totally fails to run as also in $49 \times 49$. The performance of SymVIN shows the need for more iteration: $K_{\text{layer}} = 50$ is better than $K_{\text{layer}} = 30$. However, although SymVIN can successfully run on $K_{\text{layer}} = 80$, the runs mostly fail to converge to a fixed point, which shows its limitations on stability when scaling to larger iteration. VIN does not achieve meaningful results.

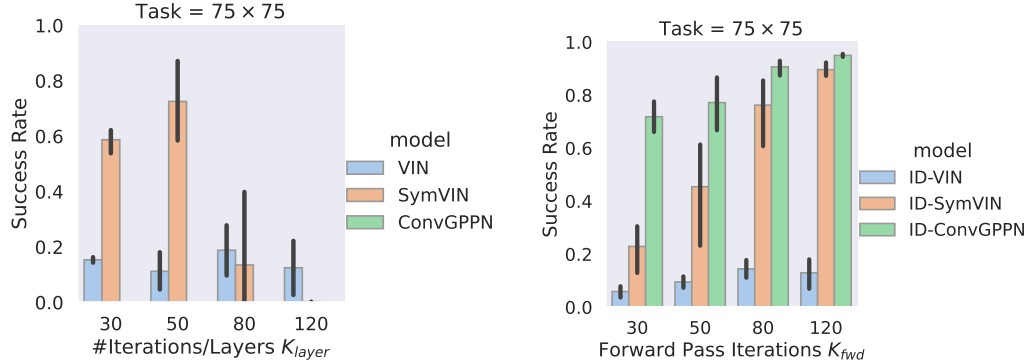

Figure 9: Performance on $75 \times 75$ maps.

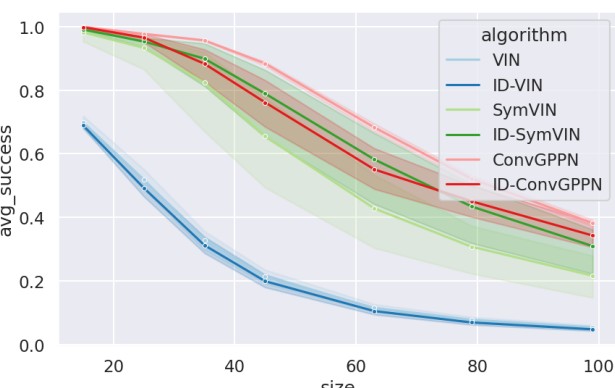

Figure 10: Generalization to larger maps.

As for implicit differentiable planners, noticeably, ID-ConvGPPN can successfully run with all forward iterations $K_{\text{fwd}}$, which is quite impressive considering its expensive recurrent network architecture. It can achieve almost perfect $100\%$ successful rate when using $K_{\text{fwd}} = 120$. For ID-SymVIN, it is less stable but can perform better with more forward iteration $K_{\text{fwd}}$, and the best performance at $K_{\text{fwd}} = 80$ or $120$ is higher than SymVIN. It also shows the potential to achieve even better number. ID-VIN seems struggling and does not give meaningful success rate.

### E.2 GENERALIZATION TO LARGER MAPS

**Setup.** In other experiments, we train the planners on the *same* map size with training case, while this experiment aims for generalization to *larger* maps to examine its potential. All methods are trained on $15 \times 15$ maps and tested on larger maps. We chose ADPs with $K_{\text{layer}} = 50$ (for stability concern) and IDPs with $K_{\text{fwd}} = 80$. All methods are tested on six sampled map sizes in $15 \times 15$ through $99 \times 99$, averaging over 5 seeds (5 model checkpoints) for each method and 1000 unseen maps for each map size. At test time, we keep the same iteration numbers as training and do not increase them. The results are shown in Figure 10.

**Results.** VIN and ID-VIN both suffer from generalizing to larger maps and perform pretty similar. ID-SymVIN is much better than ID-VIN, and outperform the explicit counterpart SymVIN. ID-ConvGPPN generalizes fine, but is worse than ConvGPPN. We find that although ConvGPPN suffers from training on large tasks (backward pass), since here we chose the best hyperparameter for ConvGPPN on $15 \times 15$, its inference (only forward pass) is pretty reliable as long as it can successfully train. As expected, the success rate of all methods drops with increasing test map sizes. While in general, both IDPs and ADPs generalize similarly well. This is expected because implicit differentiation majorly improves scalability of the backward pass, while the main bottleneck for generalization is the forward pass, where they do not have major difference.

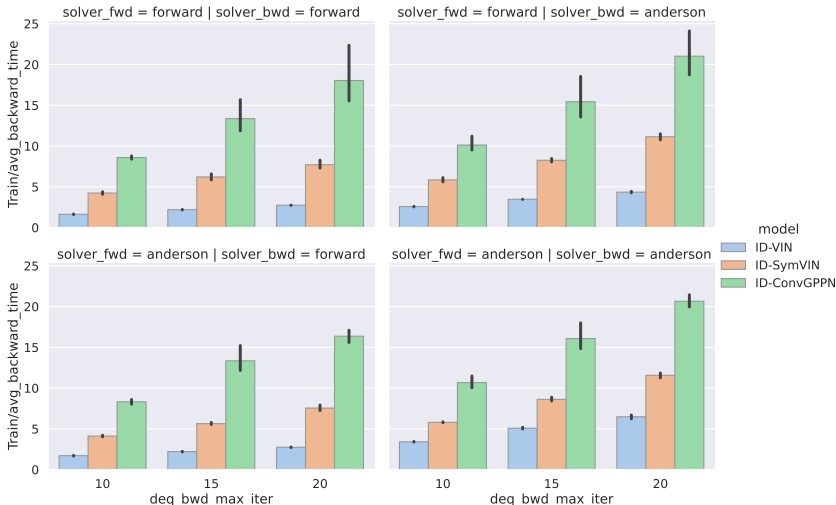

Figure 11: Backward pass iterations vs. backward runtime.

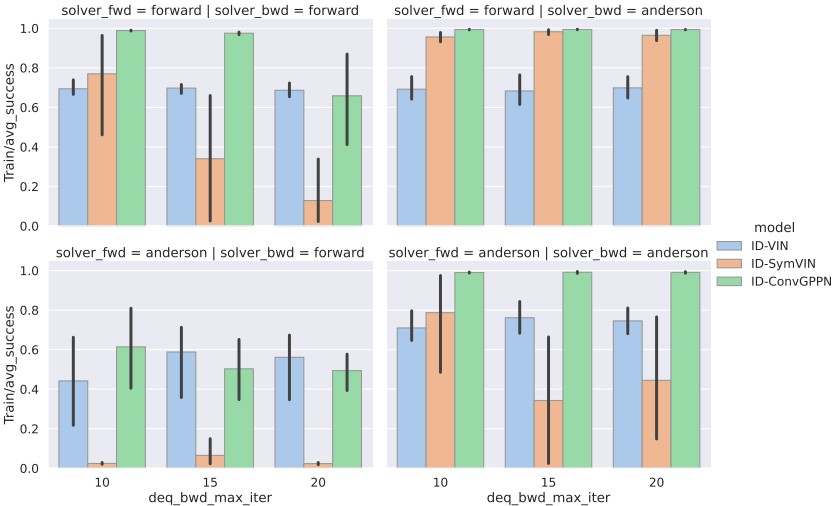

Figure 12: Success rate vs. different forward and backward solver and different backward pass iterations.

### E.3 RUNTIME OF IMPLICIT DIFFERENTIABLE PLANNERS

**Backward runtime.** We visualize the runtime of **backward pass** of IDPs for using forward solver and Anderson solver in Figure 11. The experiment is done on $15 \times 15$ maps.

As expected, using more backward pass iterations would increase the backward pass runtime. However, we also find that the backward pass has already converged at around 10 iterations, so increasing the iteration will not help the training. Instead, the iterations of the forward pass are the main bottleneck for scalability on larger maps. We show the results in the next paragraph.

**Choice of fixed-point solver.** We show the performance difference when using different solvers on all implicit differentiable planners in Figure 12. For backward passes, the Anderson solver is clearly better than the forward iteration solver. In terms of the forward passes, ID-SymVIN is not compatible with the original Anderson solver, since SymVIN needs to keep the equivariance of the intermediate variables by ordering them in a specific way. However, the Anderson solver has reshaping operations that would break it.

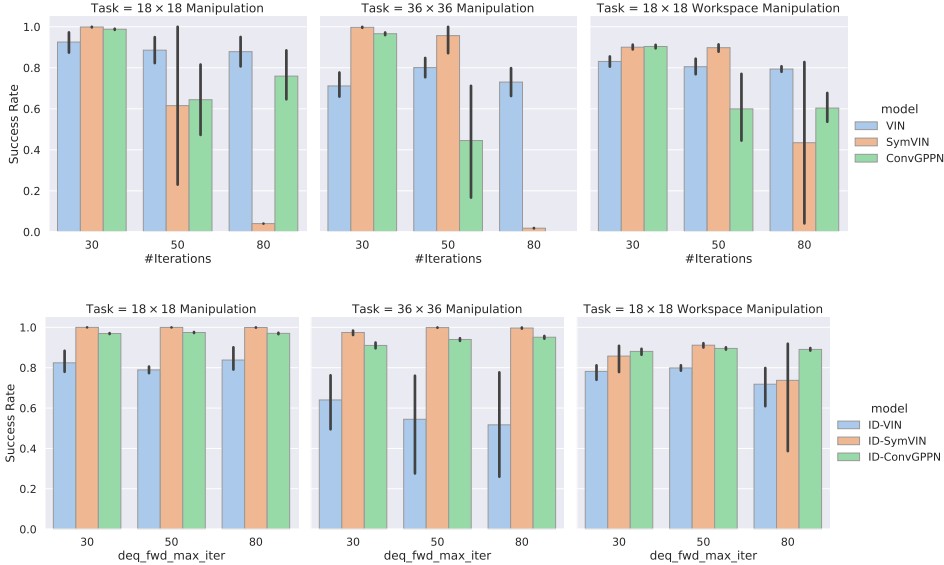

Figure 13: Manipulation complete results.

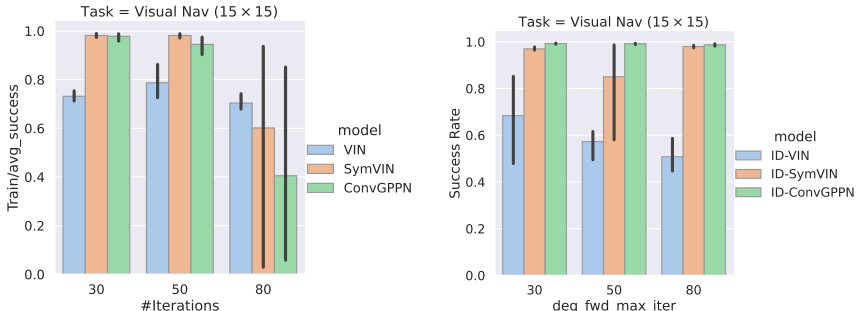

Figure 14: Visual navigation complete reuslts.

### E.4    PERFORMANCE ON MORE TASKS: COMPLETE RESULTS

In the main paper, we average over 30/50/80 forward iterations / layers. We here show the complete results for each forward iteration / layer number for manipulation in Figure 13 and for visual navigation in Figure 14. The results still follow the trends in the paper, where IDPs tend to converge more stably for larger iterations.

### E.5    JACOBIAN REGULARIZATION

We tune the Jacobian regularization from (Bai et al., 2021). We focus on tuning the Jacobian regularization weight and frequency on $15 \times 15$ maps. The x-axis is a hyperparameter of the Anderson solver for backward pass.

The results are in Figure 15. Each column corresponds to the frequency $= 0\%, 20\%, 40\%$. Each row is the weight $= 2, 4, 8$. However, the top left panel performs the best, which means zero regularization.

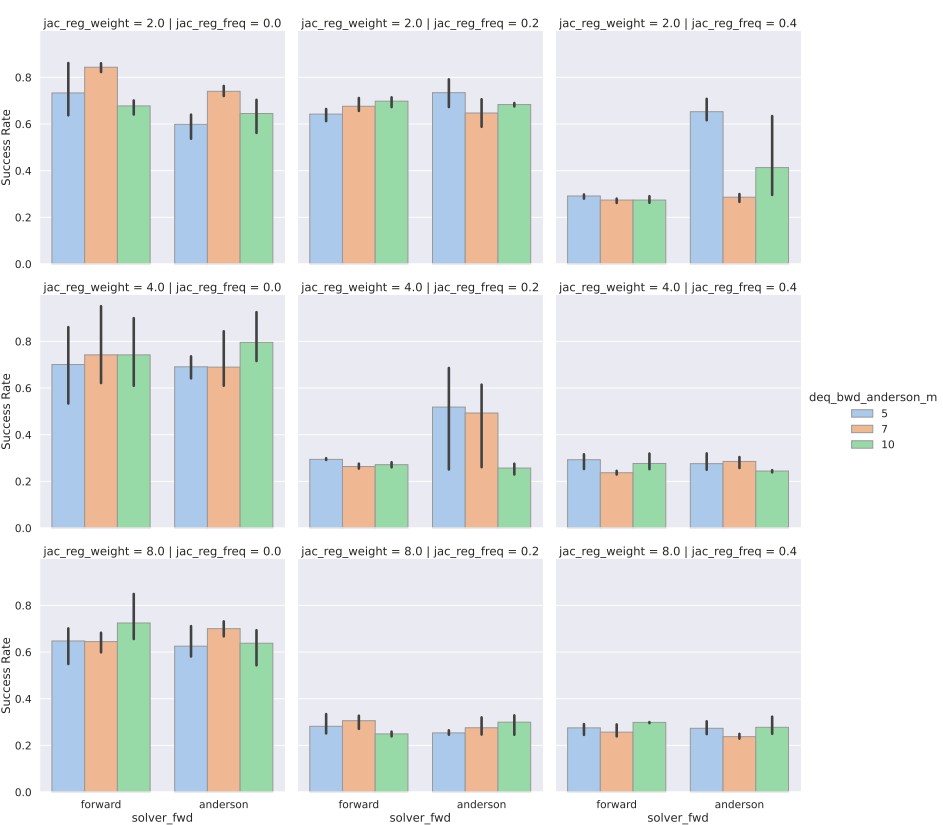

Figure 15: Tuning Jacobian regularization.

