# OpenReview forum: "Scaling up and Stabilizing Differentiable Planning with Implicit Differentiation"
_ICLR.cc/2023/Conference — ICLR 2023 poster_

### Official Review · Reviewer_Wf8g · 2022-10-25

**Confidence:** 2
**Clarity, Quality, Novelty And Reproducibility:** The paper is clear, well-written, and…
**Correctness:** 4
**Technical Novelty And Significance:** 4
**Empirical Novelty And Significance:** 4
**Recommendation:** 8

**Strength And Weaknesses:**

Strengths
+ The idea of implicitly differentiating planners that compute a fixed point is appealing
+ The implicit planners on the 2d navigation tasks in Figures 4, 6 and 7 and Table 1 convincingly improve upon the performance in contrast to the corresponding planners trained without implicit differentiation

Weaknesses
+ Sometimes the value iterations may take a long time to reach a fixed point. Is it hard in practice to ensure that a fixed-point is reaches so that the implicit derivatives remain stable?

**Summary Of The Paper:**

This paper proposes to scale differentiable planning such as Value Iteration Networks (VINs) with implicit differentiation. Section 3 summarizes value iteration networks, symmetric VINs, and gated path planning networks and how the fixed point of the value iteration can be seen as an implicit function, i.e. Bellman optimality provides an implicit function. Section 4.1 goes on to review how to compute the implicit derivatives or approximations to them, and Section 4.2 instantiates the full pipeline for implicit planning.

**Summary Of The Review:**

I recommend to accept this paper as it combines a reasonable core contribution of implicitly differentiating planners along with extensive experimental demonstrations of how it enables them to scale.

---

> ### Author Response · Authors · 2022-11-18
> **Response to Reviewer Wf8g**
>
> We appreciate the reviewer for the time and effort spent reviewing our work, as well as the support of our work. We address the concerns through individual responses and also additional results in the updated paper. We are happy to answer more concrete concerns the reviewer might have.
>
> Sometimes the value iterations may take a long time to reach a fixed point. Is it hard in practice **to ensure that a fixed-point is reaches so that the implicit derivatives remain stable**?
>
> - We did tune the planners empirically and found that unconverged fixed point in the forward pass would harm the performance.
> - We found the implicit version is actually easier to tune in terms of reaching a fixed point. Because the implicit version decouples forward and backward passes, we can increase the forward pass iteration number to guarantee it reaches a fixed point. For the explicit version (or algorithmic differentiation), we need to balance the time and memory use of the backward pass.
> - Additionally, we found as long as the implicit differentiable planners reach good fixed points in the forward pass, the backward pass only needs $K_\text{bwd}$ around 10-15 iterations across all map sizes and methods (we use $K_\text{bwd} =15$ for all cases), which implies it is not hard in practice and in other problems.
> - More concretely, we use Figures 2 and 4 to show how both types of approaches react to different forward iterations, where implicit methods are much more robust.

---

### Official Review · Reviewer_3dQr · 2022-10-26

**Confidence:** 2
**Correctness:** 3
**Technical Novelty And Significance:** 2
**Empirical Novelty And Significance:** 2
**Recommendation:** 6

**Clarity, Quality, Novelty And Reproducibility:**

This paper is clear and easy to follow. Experiments can serve as proof of concept but lack larger-scale tests. The authors provide their code. The building blocks of this pipeline are simple and should be reproducible.


**Strength And Weaknesses:**

Strengths

DEQ is suitable with the VIN-based path planning framework. The forward iteration of VIN can be viewed as solving a fixed-point equation, which is exactly what DEQ should deal with. The back-propagation of DEQ only relies on the fixed point itself, so independent of the forward solving. As a result, we can have a longer forward iteration to solve for a better convergence in some larger-scale cases.

Weaknesses

Although this paper chooses an appropriate model to tackle the VIN-based path finding problems, the technical novelty itself is somehow limited. Both the forward network structures and backward implicit differentiation schemes are from existing models.

Most importantly, the experiments cannot fully back up this paper’s main claim, that this method can scale up the path planning algorithms. The biggest map size is only 49x49, which is even less than the 100x100 examples used in the baseline SymVIN [1]. To validate the advantages of implicit differentiation, a bigger map size (with which previous baselines cannot deal) might be needed.

[1] Integrating Symmetry into Differentiable Planning



**Summary Of The Paper:**

This paper applied the implicit differentiation to the equilibrium state of the VIN-based path planning pipelines. Since the forward computation of VIN is effectively solving a fixed point of the Bellman equation, it is natural to treat this process as a deep equilibrium model (DEQ Bai et al. 2019), which can backpropagate the gradients without storing the entire forward computational graph.

**Summary Of The Review:**

In summary, this paper could be a good application of DEQ. However, the technical novelties are limited. Most of the components are from existing work. The experiments are on small scales and not compelling enough to back up the claimed scalability.

---

> ### Author Response · Authors · 2022-11-18
> **Response to Reviewer 3dQr**
>
> We appreciate the reviewer for the time and effort spent reviewing our work. We address the concerns through individual responses and also additional results in the updated paper. We are happy to answer more concrete concerns the reviewer might have, such as "correctness (3: Some of the paper’s claims have minor issues. A few statements are not well-supported, or require small changes to be made correct)" and "technical/empirical novelty (2: The contributions are only marginally significant or novel.)".
>
>
> (Concern: technical novelty) **Although this paper chooses an appropriate model to tackle the VIN-based path finding problems, the technical novelty itself is somehow limited.** Both the forward network structures and backward implicit differentiation schemes **are from existing models.**
>
> - Focus:
>     - We would like to emphasize the focus of our paper. We are problem-driven: focus on differentiable planning and identify the key problem (differentiating through a long computation path would cause scalability and stability issues), but not method-driven as in DEQ.
>     As the title suggests, the focus of our paper is “scaling up and stabilizing” differentiable planning algorithms.
> - Technical contributions:
>     - We focus on solving the issue specific to differentiable planning (more specifically, value iteration-based fixed-point solving methods)
>     - It is true that this is related to the idea of DEQ, but DEQ has a completely different motivation: it tries to extend current neural networks from feedforward (no fixed-point) to infinite-layer (with fixed-point).
>     - Furthermore, using implicit differentiation was completely new and not originally from DEQ.
>         - DEQ is also based on prior work and makes implicit differentiation work in the supervised learning domain with CNNs and Transformers.
>         - For example, implicit differentiation has been used in automatic differentiation from Rust 1998. We have specifically shown the connection in Section 4.3 (first paragraph) and Section B.1 for this.
>             - John Rust. Maximum likelihood estimation of discrete control processes. SIAM journal on control and optimization, 26(5):1006–1024, 1988.
>         - We also base our contributions on prior work using implicit differentiation, not only DEQ
>     - Using implicit differentiation here has a direct correspondence to the differentiable planning algorithm (as visualized in Figure 3) and a more interpretable limitation that is not generally true for DEQ. For example, forward steps mean iteration number.
>     - Thus, we can present more study specific to VIN and variants, not necessarily for DEQ in general. We found our case requires **different tuning and techniques, which is not trivial**. **We conclude some technical differences and novelties as follows:**
>         - One of the main technical novelties of DEQ is that it adds weight-tied and input inject, while this has already been the case in vanilla VIN.
>         - DEQ only needs around 10 forward iterations and around 5 backward iterations. We use 30, 50, and 80 forward iterations across different tasks and find 10-15 backward iterations. This is much
>         - We are learning the most important quantity in RL through the fixed-point iteration: value function. It needs great precision since the wrong relative order between Q-values $Q(s, a)$ would result in wrong actions/policies and accumulate in the entire planning procedure. However, DEQ is designed for supervised learning and does not need anything more than extracting features for input.
>         - We tried the Jacobian regularization technique (to stabilize DEQ) and other tricks/techniques that are shown useful for DEQ, but find they do not work well for path planning. This also indicates our different problem structure from theirs.
>         - DEQ and its follow-up work are still 3x-5x slower than the non-fixed-point version because it converts the original problem into solving a fixed point. However, VIN is to solve fixed point equations but requires expensive explicit differentiation (algorithmic differentiation), so implicit differentiation actually is **more scalable**. We look into this through the backward runtime and other studies.

---

> ### Author Response · Authors · 2022-11-18
> **Response to Reviewer 3dQr (part 2)**
>
> (Concern: scalability claim) Most importantly, the experiments **cannot fully back up this paper’s main claim**, **that this method can scale up the path planning algorithms**. The biggest map size is only 49x49, which is **even less than** the 100x100 examples used in the baseline SymVIN [1]. To validate the advantages of implicit differentiation, a bigger map size (**with which previous baselines cannot deal**) **might be needed**.
>
> - Previous work:
>     - We exactly follow the same setup in the paper [1] and their SymVIN training: we use $49 \times 49$ for training (they use $50 \times 50$) and also $100\times 100$ for only generalization evaluation (in Section E.1 Figure 9). In prior work, GPPN uses only $28 \times 28$ and SPT uses $50 \times 50$ (which mainly emphasizes long-term planning) for both training and evaluation.
>     - Per the request of the reviewer, we decided to add additional experiments using even larger maps for **training: $75 \times 75$, larger than all prior work.** This shows better scalability of the implicit differentiable planners.
>     - **The results are available in new Section B.1 in the revision.** We train explicit differentiable planners with 30,50,80,120 layers, and our implicit differentiable planners with max 30,50,80,120 forward pass iterations and 15 backward pass iterations. Our ID-ConvGPPN can successfully run with all iterations and achieve almost perfect results, while explicit ConvGPPN can’t run at all even at 30 layers. Our ID-SymVIN achieves a higher success rate when increasing the iteration, and the best success rate is higher than the best SymVIN number.
> - Map choice:
>     - We choose this because we follow the prior work (as pointed out, in VIN, GPPN, SPT, and so on). Specifically, VIN mainly experimented on 15x15, and GPPN mainly used 15x15 and tried 28x28 as a scalability experiment. SPT is advertised to be much better scalable with Transformers and used up to 50x50.
>     - We also have the visual navigation experiment, where we train the localization/mapping module end-to-end, which also shows scalability and versatility.
>     - Additionally, our differentiable planning module is meant to be used in a larger system. For example, a complete navigation pipeline usually composes of a global planner and a local controller. Our module can be plugged into either the global or local policy.

---

> > ### Comment · Reviewer_3dQr · 2022-12-01
> > **Thanks for your response!**
> >
> > Thanks very much for the authors' response. The 75x75 experiment indeed gives some evidence for its scalability. However, it is still not very convincing to me. In my opinion, given that one of the main claims of this paper is scalability (which is also the first word of its title), the paper would be much stronger if the authors can show even bigger map sizes (e.g. up to 500x500) so that we can see whether this method is consistently better than other algorithms, as the problem scales up. This is just a suggestion and not necessary, so I will keep my rating.

---

> > > ### Author Response · Authors · 2022-12-14
> > > **Thank you for reply!**
> > >
> > > We thank the reviewer for acknowledging our response and additional results! We understand the concern of the reviewer on further scalability. We will consider this and are happy to further investigate on even larger scale maps. Still, we would also like to kindly remind the reviewer that we focus on improving the scalability of ***training*** map sizes, but not ***inference***.
> > >
> > > In contrast, there is much work on scaling up the inference time map sizes. For example, Active Neural SLAM (Chaplot et al. 2020) uses a hierarchical paradigm by a global plus local planner on around 500x500 maps, where each planner only plans up to smaller steps.
> > >
> > > Instead, our implicit differentiable planners are more like building blocks for downstream usage. It may be possible to also use our differentiable planners in a similar hierarchical manner at inference time and potentially plan on much larger maps.

---

### Official Review · Reviewer_D5nz · 2022-10-27

**Confidence:** 5
**Correctness:** 3
**Technical Novelty And Significance:** 3
**Empirical Novelty And Significance:** 2
**Recommendation:** 6

**Clarity, Quality, Novelty And Reproducibility:**

The paper is well-written. Source code is not available (though a release is planned). There is an appendix with ample implementation details. I have some issue with the presentation and the results, which I detailed under "weaknesses".

**Strength And Weaknesses:**

**strengths**

* The paper identifies and fixes an issue with an existing method. Value iteration networks do not scale to high numbers of planning iterations. Implicit differentiation reduces the cost of the gradient computation, which allows using many more planning steps.
* Experiments show that implicit gradients allow using more planning iterations for the same cost of gradient computation. The authors also show that successful models can be trained with very high numbers of planning iterations using implicit gradients, while the same is not true of explicit ones.

**weaknesses**

I have some concerns about the results and the presentation. I will make an overview of the experiments and how I interpret the results below to make that clear.

**overview of experiments**

The paper has 4 experiment setups: 1) 2D navigation 2) visual navigation 3) configuration-space manipulation 4) work-space manipulation.

**2D navigation**

There are maps of size 15x15, 27x27 and 49x49. According to Figure 2, in both 15x15 and 27x27 there exist explicit planners which reach 100% success. In 49x49 maps, the best explicit planner (SymVIN with 30 iters) reaches about 85% success.

According to Figure 4, the implicit planners match the performance of explicit ones in all scenarios, and even exceed that in 49x49 with 80 iterations using ConvGPPN. That being said, according to Figure 6 the runtimes of implicit planners in the 49x49 task seem impractical for all models (except VIN, which performs poorly).

To sum up, implicit vs explicit perform roughly on par. There are setups where implicit works better, but the runtime gets quite high.

**visual navigation**

According to table 1, implicit methods clearly outperform explicit ones. However, table 1 averages results across 30, 50 and 80 iterations. We can find individual runs in Figure 13. Here, the best explicit methods are SymVIN and ConvGPPN with 30 iterations, which reach ~100% success. The implicit ones also reach this performance. To sum up, implicit vs explicit perform on par.

**configuration-space manipulation**

There are 18x18 and 36x36 setups. Again, table 1 shows implicit methods outperforming the others on averaged results. Figure 11 contains individual results for 30, 50 and 80 iterations. In both 18x18 and 36x36, the best explicit method (SymVIN with 30 iters) reaches ~100% success rate. Implicit methods also reach this performance. So again, the two are on par.

**work-space manipulation**

Similar to configuration space, averaged results favor implicit methods. According to Figure 11, the best explicit methods are SymVIN and ConvGPPN with 30 iterations, and they reach between 85 and 90% success. The best implicit method seems to be SymVIN with 50 iterations. This performs similar to the explicit ones, but perhaps with a slight edge. Without exact numbers I'd have to say the best explicit and the best implicit perform on par.

---

Looking at these results, I have two main concerns:

* Implicit and explicit methods appear to perform on par, if we compare the best explicit method for each setup with the best implicit method. The only setup where implicit has an edge is the 49x49 2D navigation task. However the runtimes of the implicit method here appear impractical. The backward pass of SymVIN takes 50 seconds according to Figure 6, which is quite long. That would mean over an hour for 100 gradient updates.
* I find the presentation a bit misleading at times. In Table 1, it doesn't make sense to me to average across planning iterations. A fair comparison would be best-vs-best. Also, I find that there is not enough discussion of runtimes. I greatly appreciate Figure 6 but I still think more space should be dedicated to discussing runtimes. The results don't seem to be as simple as "implicit gradient is faster". There are some settings where the implicit gradient is slower than the explicit one (e.g. 49x49 SymVIN with 30 iterations) and settings where the fact that the implicit gradient is faster doesn't matter too much because it is still not fast enough (e.g. 49x49 ID-SymVIN backward pass takes close to a minute). A fair comparison would also include the runtime of the best implicit method against that of the best explicit one.

Please let me know if I'm misinterpreting the results here.

My recommendation would be to point out a setup where a) the implicit version is feasible, while the explicit is not b) the implicit version has an edge over the explicit one. By b) I mean that the longer planning horizon allows the method to reach a level of performance that an explicit gradient cannot reach with any number of iterations that is applicable or any choice of model. This would make it easier for the reader to understand when implicit gradients are helpful. I have searched for such a setup in the paper but couldn't find it (see my process above under "overview of experiments").

**some minor points**

* Can the authors define what they mean by divergence? Do you mean that the gradient vanishes or explodes?
* One missing related work that I spotted: [1]. It has a similar goal as this paper: scaling up planning to larger environments.

[1] Value Iteration Networks on Multiple Levels of Abstraction, http://www.roboticsproceedings.org/rss15/p14.pdf

**Summary Of The Paper:**

The paper proposes to use implicit differentiation in value iteration networks, allowing for a deeper network structure/more iterations of planning. This is because the implicit gradient has a constant complexity with respect to the number of planning iterations.

**Summary Of The Review:**

The paper proposes using implicit differentiation in value iteration networks (VINs). This allows using many more planning iterations than usual. The authors show that they can successfully train VINs with very high numbers of planning iterations, which is not possible using explicit gradients. I generally do not see a performance gap between implicit and explicit gradients in the experiments. The few cases where there is a benefit for implicit methods seem to also come with a very high runtime. The presentation does not make this very clear due to the reporting of averaged results. I find the idea of using implicit differentation in VINs very promising and interesting, but the experimental results and presentation currently do not convince me. The paper can be made a lot stronger by including experiments which show implicit methods clearly outperforming explicit ones and with reasonable runtimes. At the current stage, I see the paper as below the threshold of acceptance.

---

> ### Author Response · Authors · 2022-11-18
> **Response to Reviewer D5nz (part 1)**
>
> We appreciate the reviewer for taking precious time on writing such detailed comments. We found lots of them insightful and helpful for improving the paper.
>
> We will respond to all concerns correspondingly (summarized by points below), where some  We believe there are some misunderstandings being made unconsciously and will try to clarify them. We also add additional experiments and analyses that can provide more insight to better support our claims, which the reviewer may find useful.
>
> We are happy to answer more concrete concerns the reviewer might have, such as "correctness (3: Some of the paper’s claims have minor issues. A few statements are not well-supported, or require small changes to be made correct)" and "technical/empirical novelty (2: The contributions are only marginally significant or novel.)".
>
>
> **(Minor concern 1)** **Can the authors define what they mean by divergence? Do you mean that the gradient vanishes or explodes?**
>
> - The term “divergence” can indeed be confusing. We answer this first since this directly relates to the understanding of the main messages of the paper.
> - In value iteration, the goal of the **forward pass** is to find a fixed point by iteratively applying the Bellman optimality operator. For an MDP with a **********given********** dynamics model, Bellman (optimality) operators induced from the given model and arbitrary policy (when discount factor < 1) are guaranteed to be a contraction mapping, so iterative methods will provably converge to a unique fixed point.
> - However, the entire network is learned end-to-end, and the transition weights are also learned, thus this property of Bellman operators is not guaranteed. When the planning horizon increases, this issue becomes more severe since it can cause the iteration even farther from the fixed point.
> - When we talk about divergence, we mainly mean this fixed-point iteration in the **forward pass**. This iterative process **does not** involve gradient computation at all, but is purely the property of VIN and variants.
> - Additionally, implicit differentiable planners also introduce a fixed-point iteration for the backward pass. Empirically, it does not have a divergence issue.
>
> **(Part of concern 1)** **Implicit differentiable planners can successfully run on 49x49 2D navigation, but the runtime is impractical?**
>
> - We would like to clarify the runtime here and apologize for any confusion. Figure 6 shows the runtime for an epoch of gradient descent over the entire dataset. We have 10K maps in training and use a batch size of 32, so a total $10000/32 \approx 313$ gradient updates. Thus the 100 seconds for a backward pass means just around 0.3 seconds for each update, similarly for a forward pass.
> - In total, we use 60 epochs for all sizes, so the training (and additional evaluation) may take from less than one hour to a few hours, but up to around 10 hours for ID-ConvGPPN on the largest tasks.
> - From common sense, our planner is just a convolution network with $K_\text{layer}$ layers (which can take e.g. 30, 50, or 80), so training such a network has no reason to take so long and will not be impractical.

---

> ### Author Response · Authors · 2022-11-18
> **Response to Reviewer D5nz (part 2)**
>
> **(Main concern 1: performance)** **Implicit and explicit planners perform on par on all other environments (2D nav smaller maps, visual nav, C-space manipulation, workspace manipulation)? / “**My **recommendation** would be to **point out a setup where a) the implicit version is feasible, while the explicit is not b) the implicit version has an edge over the explicit one.** **”**
>
> - We would like to emphasize the central research question of the paper. As the title suggests, we want to **scale up** and ******************stabilize****************** the **training** of differentiable planning algorithms.
> - (Our focus) Because VIN and its variants are computing the fixed point to the Bellman equation, we consider the **central ability** is its **convergence property** to the fixed point.
>     - Thus, by **scaling up**, we specifically refer to **successfully training with larger iterations**. By **stabilizing**, we specifically refer to successfully **converging to the fixed point**.
>     - Empirically, we found **the key technical difficulty in these methods is using larger iterations**, since they decide the **upper bound** of the **performance**. Using larger tasks would require more iterations, but it is **not** the fundamental cause. This is why we consistently compare all methods and all tasks with different forward iterations.
> - (Performance wise) We **do not claim** that using implicit differentiation itself **should yield better performance**, because implicit differentiation is equivalent to explicit differentiation (algorithmic differentiation) in **asymptotic** performance. They are **not supposed to performance differently**, if we can **run long enough**. However, the implicit differentiable planners **scale up better** since they can run **use fewer resources, while the explicit ones can’t keep up the scale.**
> - (Our interpretation) Given this focus, we mainly compare implicit vs. explicit in Figures 2 and 4 for 2D navigation. We empirically show that implicit ones can scale up to larger iterations (run successfully) and can stably converge for larger iterations.
>     - For other environments, we also use different iterations to understand the upper bound of implicit and explicit approaches.
> - (Additional experiment) **Per request of the reviewer, we decided to add additional experiments using even larger maps for training: $75 \times 75$. This shows why more iteration is useful.**
>     - Prior work GPPN uses only $28 \times 28$ and SPT uses $50 \times 50$ (which mainly emphasizes long-term planning) for both training and evaluation. Note that SymVIN also only uses $50 \times 50$ for training, while using up to $100 \times 100$ for generalization. We also did the same experiment, available in Section E.1.
>     - **The results are available in new Section B.1 in the revision.** We train explicit differentiable planners with 30,50,80,120 layers, and our implicit differentiable planners with max 30,50,80,120 forward pass iterations and 15 backward pass iterations. Our ID-ConvGPPN can successfully run with all iterations and achieve almost perfect results, while explicit ConvGPPN can’t run at all even at 30 layers. Our ID-SymVIN achieves a higher success rate when increasing the iteration, and the best success rate is higher than the best SymVIN number.
>
> **(Main concern 2: presentation)** “I find **the presentation a bit misleading at times**. In Table 1, it doesn't make sense to me to average across planning iterations. **A fair comparison would be best-vs-best**.”
>
> - We did have a discussion on whether to use best-vs-best or averaging over all planning iterations. We finally decided to average over planning iterations because we care about stability. We show complete results in appendix Section E.3 and did not mean to hide anything.
>
> **(Main concern 2, continued: runtime)** “**Also, I find that there is not enough discussion of runtimes.” / “There are some settings where the implicit gradient is slower than the explicit one” / “A fair comparison would also include the runtime of the best implicit method against that of the best explicit one."**
>
> - The research question of our paper is the scalability and stability of value iteration convergence using different iterations. Thus, we think it is more meaningful to compare runtime separately for each iteration, instead of comparing runtime based on performance (e.g. best vs best).
> - We believe that would make the figure harder to interpret. Especially, we mainly want to compare large tasks/iterations, such as 80 iterations or 49x49, where explicit methods struggle.
>
> **(Minor concern 2)** **One missing related work that I spotted: [1]. It has a similar goal as this paper: scaling up planning to larger environments.**
>
> - Thank you for suggesting the work, we added it to the revision.

---

> ### Comment · Reviewer_D5nz · 2022-11-23
> **Thank you for your response**
>
> Thank you for your response. Unfortunately, I still believe that the paper is below the threshold of acceptance.
>
> My main reason is that the paper does not provide enough motivation for why it is useful to be able to do more iterations. Currently, the only experiment where doing more iterations actually brings a performance improvement is in the appendix (the 75x75 navigation task). For all of the experiments in the main paper, a method with explicit gradients and a relatively low number of iterations is sufficient for good performance. I see this as a problem because implicit differentiation has its own drawbacks: it is actually slower in the low-iteration regime, according to Figure 6. I also am not convinced that Table 1 should present averaged results. If the point is to show that the explicit methods start to fail with larger numbers of iterations, this should be shown in a more deliberate way. Simply averaging across suboptimal hyper-parameters of a baseline gives a misleading sense of its performance.
>
> I do believe that this work is promising and would strongly encourage the authors to continue this line of research. However, I would recommend focusing more on experiments like the 75x75 navigation task, which actually showcases a problem where it is necessary to use many more iterations, which is only possible through implicit differentiation. Gathering a wider set of such cases would make this paper much stronger.

---

> > ### Author Response · Authors · 2022-11-24
> > **Authors' Response; Thank you for additional feedback**
> >
> > We thank the reviewer for further feedback on our paper and response, and we also appreciate the reviewer acknowledging our line of research! We believe the discussion has helped us make the claims and messages more precise. We respond to two concerns and hope to give a more concise account and explain our reasoning.
> >
> >
> > > “**My main reason is that the paper does not provide enough motivation for why it is useful to be able to do more iterations” / “I would recommend focusing more on experiments like the 75x75 navigation task, which actually showcases a problem where it is necessary to use many more iterations, which is only possible through implicit differentiation**.”
> >
> > - We would like to emphasize again that our focus is on the scalability and stability of the differentiable planning algorithms, not on achieving higher numbers on **all** tasks. Among all aspects, we found the major cause of scalability and stability issues of differentiable planning is when doing more iterations.
> >     - We think the experiment on 49x49 2D navigation already provides enough **motivation**. When we use more tasks, we aim to show **additional empirical evidence** that explicit differentiable planners struggle with scaling up and stabilizing convergence.
> > - For the comment on “**I would recommend focusing more on experiments like the 75x75 navigation task**”, we do agree on the importance of large-scale tasks. The reason we showed tasks in all scales is intended to provide more comprehensive study, because we **did not** claim that implicit differentiable planners are always better, but mainly for large-scale iterations/tasks.
> >     - With 2D maze navigation in 15x15, 27x27, and 49x49, we want to show that our implicit differentiable planners are not only beneficial on large-scale iterations/tasks but also comparable for medium-scale or small-scale tasks.
> >     - This does not come with guarantees and needs evidence from experiments, because less iteration would cause worse fixed points for implicit differentiation and degenerate performance.
> >     - We eventually find there is a clear boundary on when to use implicit or explicit differentiable planners, detailed in the last paragraph of Section 4.3 and the conclusion in Section 6.
> >
> > > **I also am not convinced that Table 1 should present averaged results.**
> >
> > - We have considered this “best-vs-best” presentation, although eventually, we chose another one. We respect the preference of the reviewer and did not mean to convince the reviewer of this. Here we summarize our thoughts on this.
> > - We think the disagreement may be also related to the understanding of the paper's focus. Our focus is scalability and stability, not achieving higher numbers on **every** task. We find “best-vs-best” for each task may not highlight the stability/scalability aspects.
> >     - In other words, explicit differentiable planners should have performed better with more iteration, but it failed to do so for some iterations, while best-vs-best cannot highlight this. We thus chose the "averaged" results.
> > - We do not find either “best-vs-best” or averaged to be the perfect presentation. In the original paper, to make up for this, we included complete results in the appendix Section E.3 (separate out success rate for each iteration, Section F.3 in the revised version).
> > - Thus we consider this not as a drawback or a misleading presentation since we already present the complete results in the original paper and our reasoning in the rebuttal.

---

> ### Comment · Reviewer_D5nz · 2022-12-06
> **Revising my score**
>
> Revisiting this paper some time later, I believe I was too harsh in my first assessment. I think showing that implicit differentiation can be used to train value iteration networks with many more iterations/layers is a worthy contribution in itself. I will increase my score from 5 to 6 because of that. I still find that the set of problems where we see a benefit from ID is too narrow. In the end, the ability to do more iterations itself is not what we are after. We are after tangible benefits (e.g. faster operation, better performance, etc.).
>
> I also still find Table 1 to be poorly designed. I see the authors' argument, that they want to show that explicit differentiation fails with more iterations/layers, but I find that showing averaged results sends the wrong message. Of course, the reader can obtain the full picture using the rest of the paper, but ideally a table or figure should be understandable on its own. When you average results across hyper-parameters in a table, the reader doesn't know if a score is low because of outliers or because it systematically under-performs. My point is not that you should make a best-vs-best comparison. I will repeat what I wrote in my second comment here:
>
> >  If the point is to show that the explicit methods start to fail with larger numbers of iterations, this should be shown in a more deliberate way.
>
> To summarise, I believe the paper is slightly over the threshold. For a higher score, I would have to see a wider set of problems where implicit differentiation has an edge over explicit differentiation. Also, the presentation issues related to Table 1 would have to be solved.

---

> > ### Author Response · Authors · 2022-12-14
> > **Thank you for being willing to revisit the review and score**
> >
> > We appreciate the reviewer for being willing to revisit our paper and reconsider the review! This is encouraging for us.
> > - We are keeping considering these points. Regarding the concern on Table 1, our authors had further discussions on how to further revise the table and understand that averaged case could be confusing.
> > - We have several solutions in the discussion, including (1) sacrificing some other content and moving the full figures/tables to the main paper, (2) choosing one hyperparameter for each method (such as $K_{layer}=30$ for the explicit case and $K_{fwd}=80$ for ), and (3) using best-vs-best for each cell.
> > - We are happy to take additional feedback on the choices if there are any. We also agree that in any case, we should add the best-vs-test or the fixed hyperparameter one at least to the appendix. We will make the changes to the next revision (e.g. camera ready if applicable).
> > - Additionally, we have some kind reminders:
> >     - Although we did not intend to compare $K_{layer}$ for the explicit case to $K_{fwd}$ for the implicit case (the “hyperparameter” column), we also did not have better solutions for aligning the comparison and thus compromised.
> >     - We regard Figures 2 and 4 (2D Navigation) as the main results and show complete results. We hope they show similar messages as the complete results of manipulation and visual navigation, thus we compressed Table 1.

---

### Author Response · Authors · 2022-11-18
**General response**

We thank all reviewers for their thoughtful and detailed reviews! We address the concerns in individual responses.

We added the additional experiment on training 2D navigation on $75 \times 75$ maps to Section B.1, as requested by reviewers for scalability comparison.

---

### Decision · Program_Chairs · 2023-01-20

**Decision:**

Accept: poster

**Justification For Why Not Higher Score:**

See meta-review.

**Justification For Why Not Lower Score:**

See meta-review.

**Metareview: Summary, Strengths And Weaknesses:**

The paper improves value-iteration networks by using implicit differentiation, thus allowing for scaling of these methods.  That contribution is nice, albeit that the merit of this approach would be stronger with wider experimentation (but that's true for any paper).



**Note From Pc:**

if the above contains the word "oral" or "spotlight" please see: "oral" presentation means -> notable-top-5% and "spotlight" means -> notable-top-25%. As stated in our emails, we are disassociating presentation type from AC recommendations